# Alterations in the Metabolic and Lipid Profiles Associated with Vitamin D Deficiency in Early Pregnancy

**DOI:** 10.3390/nu17193096

**Published:** 2025-09-29

**Authors:** Yiwen Qiu, Boya Wang, Nuo Xu, Shuhui Wang, Xialidan Alifu, Haoyue Cheng, Danqing Chen, Lina Yu, Hui Liu, Yunxian Yu

**Affiliations:** 1Department of Public Health and Department of Anesthesiology, Second Affiliated Hospital of Zhejiang University School of Medicine, Hangzhou 310009, China; 2Department of Epidemiology & Health Statistics, School of Public Health, School of Medicine, Zhejiang University, Hangzhou 310058, China; 3Women Hospital, School of Medicine, Zhejiang University, Hangzhou 310006, China; 4Zhejiang Key Laboratory of Pain Perception and Neuromodulation, Hangzhou 310009, China; 5Clinical Research Center, Sir Run Run Shaw Hospital, School of Medicine, Zhejiang University, Hangzhou 310016, China

**Keywords:** vitamin D deficiency, 25(OH)D, metabolomics, pregnancy, lipids

## Abstract

**Objective:** Vitamin D deficiency (VDD) is common in pregnancy and may affect lipid metabolism. The underlying mechanisms are multifactorial, but most evidence so far comes from non-pregnant populations. This study aims to identify metabolites and metabolic patterns associated with VDD in early pregnancy and to evaluate their relationships with maternal lipid profiles. **Methods:** A nested case–control research was carried out in the Zhoushan Pregnant Women Cohort (ZPWC). Cases were defined as women with VDD (25(OH)D < 20 ng/mL), and controls (≥20 ng/mL) were matched 1:1 using propensity scores based on age, pre-pregnancy BMI, gestational week, and calendar year at blood sampling. The untargeted metabolomics of first-trimester maternal plasma were measured. Metabolic profiles were analyzed using partial least squares-discriminant analysis (PLS-DA). Principal component analysis (PCA) was applied to visualize group separation, and metabolite set enrichment analysis (MSEA) was performed to reveal biologically relevant metabolic patterns. Associations between VDD-related metabolite components in early pregnancy and lipid levels in mid-pregnancy were assessed using linear regression models. **Results:** 44 cases and 44 controls were selected for the study. There were 60 metabolites identified as being connected to VDD. Among these, 26 metabolites, primarily glycerophospholipids and fatty acyls, exhibited decreased levels in the VDD group. In contrast, 34 metabolites showed increased levels, mainly comprising benzene derivatives, carboxylic acids, and organooxygen compounds. PCA based on these metabolites explained 52.8% of the total variance (R^2^X = 0.528) across the first six principal components (PC1: 16.4%, PC2: 10.6%, PC3: 9.2%, PC4: 6.3%, PC5: 5.7%, PC6: 4.6%). PC2, dominated by lineolic acids and derivatives, was negatively associated with total cholesterol (TC), triglycerides (TG), and low-density lipoprotein cholesterol (LDL-C) (all *p* < 0.01). PC3, dominated by glycerophosphocholines, was negatively associated with TC, TG, and high-density lipoprotein cholesterol (HDL-C) (all *p* < 0.05). MSEA revealed significant enrichment of the pantothenate and CoA biosynthesis pathway after multiple testing correction (FDR < 0.05). **Conclusions:** This study reveals distinct metabolic alterations linked to VDD and suggests potential mechanisms underlying its association with maternal lipid metabolism in early pregnancy.

## 1. Introduction

Vitamin D is a fat-soluble vitamin with diverse physiological functions beyond skeletal health, including roles in immune regulation and metabolic processes [1]. It has been acknowledged that vitamin D deficiency (VDD) is a widespread global public health issue, with an even higher prevalence among pregnant women [2]. The prevalence of VDD among pregnant women in China is approximately 41.96%, and we observed an even higher prevalence rate of 65.26% in coastal regions of eastern China [3,4]. Meanwhile, there is growing evidence that vitamin D status is linked to dyslipidemia in pregnant women. For example, lower levels of 25(OH)D in early pregnancy were linked to higher levels of total cholesterol (TC), low-density lipoprotein cholesterol (LDL-C), and total cholesterol to high-density lipoprotein cholesterol ratios (TC/HDL-C) during gestation, according to a prospective cohort study conducted on pregnant Brazilian women [5]. Consistent with these results, a prior study found a negative relationship between TC, triglycerides (TG), and LDL-C in mid-pregnancy and serum 25(OH)D levels in early pregnancy, which suggests that higher maternal vitamin D levels may help improve lipid profiles during pregnancy [6]. Previous research has mostly concentrated on how vitamin D’s antioxidant and anti-inflammatory mechanisms may affect lipid metabolism; however, additional underlying mechanisms remain to be explored [7].

Metabolomics examines endogenous and exogenous metabolites to generate a comprehensive view of biological systems, offering a powerful means to uncover disruptions in metabolic pathways associated with phenotypic variation and to identify novel targets for therapeutic intervention. The relationship between maternal 25(OH)D and metabolome during pregnancy has been investigated in earlier research. The U.S. study (*n* = 30), conducted in a cohort of adolescent pregnant women, found that several metabolites related to inflammation and fatty acid oxidation were decreased in the vitamin D sufficient group during mid-pregnancy [8]. In contrast, in a Chinese study (*n* = 111), based on early-pregnancy blood samples from an adult cohort, vitamin D insufficiency or deficiency women showed higher concentrations of glycerophospholipids as well as aromatic and branched-chain amino acids [9]. The Danish study identified 42 metabolites associated with maternal 25(OH)D levels at one week postpartum with the method of linear regression, with these metabolites significantly enriched in the sphingolipid subpathway [10]. Another study focusing on fatty acids reported that maternal 25(OH)D levels were positively associated with plasma omega-6 fatty acids, and total polyunsaturated fatty acids, while being inversely associated with total saturated fatty acids and total monounsaturated fatty acids at delivery [11]. In addition, a study (*n* = 80) in Turkey demonstrated that VDD was associated with altered profiles of bile acids [12]. Although these studies were conducted across diverse populations and different stages of pregnancy, they collectively suggest a potential link between vitamin D status, lipid-related metabolites, and metabolic pathways. However, no study has yet elucidated the relationship between VDD, metabolite alterations, and their potential contribution to dyslipidemia during pregnancy.

In this study, a nested case–control design was employed to conduct untargeted metabolomic profiling of maternal plasma samples collected in early pregnancy, with the aim of identifying metabolites and metabolic pathways associated with VDD and assessing their associations with maternal lipid levels.

## 2. Methods

### 2.1. Study Design and Participants

The Zhoushan Pregnant Women Cohort (ZPWC), an ongoing prospective cohort that was started in August 2011 at Zhoushan Maternal and Child Health Care Hospital in China, served as the basis for the nested case–control research design of this investigation. Eligibility for ZPWC required participants to: (1) provide informed consent; (2) be enrolled between 8 and 12 gestational weeks; (3) complete routine perinatal examinations and plan to deliver at the study hospital; (4) be between 18 and 45 years of age; (5) have no family history of psychiatric illness. Women were excluded if they had: (1) a history of major chronic or acute diseases; (2) a history of psychiatric disorders before pregnancy; (3) threatened abortion; (4) fetal malformations or abnormal fetal growth; or (5) difficulties completing questionnaires due to cognitive impairment. During the ZPWC study, trained nurses conducted in-person interviews to gather participants’ sociodemographic details, lifestyle habits, and health-related behaviors. Laboratory examination results were taken from the electronic medical record system of the hospital. Fasting blood samples in each trimester were also collected. The Zhejiang University School of Medicine’s Research Ethics Committee gave its approval to the study protocol (Approval No. 2011-1-002).

For the present analysis, we first identified all women with VDD (serum 25(OH)D < 20 ng/mL) in early stage of pregnancy from our cohort. Among these, 44 women were randomly selected as cases. Controls were drawn from participants with adequate vitamin D levels (25(OH)D ≥ 20 ng/mL) at the same gestational stage. To minimize potential confounding, cases and controls were matched 1:1 using propensity score matching based on maternal age, pre-pregnancy BMI, calendar year, and gestational week at blood sampling, yielding 44 matched pairs (*n* = 88).

### 2.2. Clinical Data Collection

#### 2.2.1. Measurement of 25(OH)D

Plasma concentrations of 25(OH)D_2_ (ng/mL) and 25(OH)D_3_ (ng/mL) were quantified using liquid chromatography–tandem mass spectrometry (LC–MS/MS) with an API 3200MD system (Applied Biosystems/MDS Sciex, Waltham, MA, USA). Plasma 25(OH)D levels (ng/mL) were obtained by summing 25(OH)D_2_ and 25(OH)D_3_.

#### 2.2.2. Measurement of Blood Lipids

Serum levels of TC, TG, HDL-C, and LDL-C in the mid-pregnancy were measured using enzymatic methods on an AU5800 automated analyzer (Beckman Coulter, Brea, CA, USA). All assays followed manufacturer protocols, with inter-assay coefficients of variation < 5%.

#### 2.2.3. Variable Definition

Self-reported height and weight data were used to calculate the maternal pre-pregnancy BMI (kg/m^2^). Reproductive history included gravidity (1, 2, or ≥3 pregnancies) and parity (nulliparous vs. multiparous). Education levels were stratified into three categories: (1) junior high school and below, (2) senior high school, and (3) college and above. Seasonal variations in vitamin D assessment were adjusted for differences in sunlight exposure: spring (March to May), summer (June to August), autumn (September to November), and winter (December to February). Missing data were coded as “unknown”.

### 2.3. LC–MS Metabolomics Data Acquisition and Preprocessing

Untargeted metabolomic analysis was performed using LC-MS/MS with a Q-Exactive series Orbitrap mass spectrometer (Thermo Fisher Scientific, Waltham, MA, USA). Raw data preprocessing included the following steps: (1) Missing Value Handling: Features with > 50% missing values in any group were removed, and the remaining entries were imputed as half of the minimum observed value for each metabolite. (2) Features showing > 90% relative standard deviation (RSD) excluded from further analysis. (3) To reduce the effects of noise and the high variance of the variables, the data were scaled and logarithmically processed. After preprocessing, 14,661 features were retained, of which 648 were retained for secondary qualitative substances.

### 2.4. Data Analysis

According to their non-normal distribution, all continuous variables were summarized with the median [IQR]. The frequency of categorical variables was displayed as a percentage. Between-group comparisons were conducted using the chi-square test for categorical data and the Mann–Whitney U test for continuous data. Fisher’s exact test was utilized if the expected frequency of any cell was below five.

Multivariate analysis was conducted using partial least squares-discriminant analysis (PLS-DA). Metabolites with variable importance in projection (VIP) scores >1.0 and statistically significant differences (*p* < 0.05) were identified as VDD-associated metabolites. These discriminant metabolites were subsequently analyzed using principal component analysis (PCA) to visualize group separations. Metabolite set enrichment analysis (MSEA) was performed using the MetaboAnalyst 6.0 platform (https://www.metaboanalyst.ca/, accessed on 30 December 2024) to identify biologically relevant metabolic patterns, in order to reveal significant enrichment of key pathways associated with vitamin D metabolism. Linear regression modeling was used to analyze the association between VDD-related metabolites and lipid levels in mid-pregnancy. The results were visualized using forest plots, with β coefficients and 95% confidence intervals representing the effect sizes.

The level of *p* < 0.05 was considered as statistically significant. All statistical analyses were performed in R software (version 4.3.2).

## 3. Results

### 3.1. Baseline Characteristics

44 participants were included in each of the control and case groups, and no significant differences were observed in baseline demographics, gestational week or season of sampling between VDD and non-VDD participants (all *p* > 0.05), while 25(OH)D and 25(OH)D_3_ levels were significantly lower in the VDD group (*p* < 0.001) (Table 1).

### 3.2. VDD and Maternal Metabolic Profile in Early Pregnancy

The score plot for multivariate modeling using the PLS-DA method is shown in Figure 1A, defining metabolites with VIP > 1 and *p* < 0.05 as VDD-associated metabolites. The model shows good goodness-of-fit (R^2^Y = 0.897) and moderate predictive ability (Q^2^Y = 0.325), as well as a low explained variance in metabolic profiles (R^2^X = 0.064). The volcano plots depicting increases and decreases in VDD-associated metabolites (N = 60) as well as their belonging to the Superclass (Figure 1B). Among the identified metabolites, 21 of them were categorized as lipid and lipid-like molecules, 11 as benzenoids, and the remaining included organic acids and derivatives, organoheterocyclic compounds, and organic oxygen compounds. At the class level, a total of 26 metabolites, mainly represented by 8 glycerophospholipids and 4 fatty acyls, were decreased in the VDD group. In contrast, 34 increased metabolites in the VDD group were primarily classified as benzene and substituted derivatives (*n* = 4), carboxylic acids and derivatives (*n* = 4), and organooxygen compounds (*n* = 4) (More details in Appendix A).

Based on VDD-associated metabolites obtained from PLS-DA, unsupervised PCA modeling captured 52.8% of the cumulative variance in the metabolic profiles across the first six principal components (PC1: 16.4%, PC2: 10.6%, PC3: 9.2%, PC4: 6.3%, PC5: 5.7%, PC6: 4.6%) (Figure 2).

Table 2 presents the metabolites that had primary influence on the top three principal components (absolute value of score > 0.2). PC1 exhibited a complex composition, dominated by benzenoids, lipids and lipid-like molecules; PC2 was dominated by lipids and lipid-like molecules, including three lineolic acids and derivatives; and PC3 was dominated by organoheterocyclic compounds, lipids and lipid-like molecules, with five glycerophosphocholines.

To explore the systemic metabolic impact of VDD, we conducted MSEA using the KEGG pathway library (Table 3). Among all evaluated pathways, only pantothenate and CoA biosynthesis (*p* = 0.0002) and unsaturated fatty acid biosynthesis (*p* = 0.0387) showed statistically significant enrichment. After multiple-test corrections, only the pantothenate and CoA biosynthesis remained significant at an FDR threshold of 5% (*p* = 0.0118).

### 3.3. VDD-Associated Metabolites and Lipid Levels

In Figure 3, linear regression analysis showed that PC2, which is dominated by lineolic acids and derivatives, was significantly inversely related to TC (β = −0.130, 95%CI: −0.201, −0.059), TG (β = −0.063, 95%CI: −0.100, −0.026) and LDL-C (β = −0.081, 95%CI: −0.136, −0.026). Conversely, PC3 mainly composed of glycerophosphocholines showed significant negative correlation with TC (β = −0.097, 95%CI: −0.175, −0.019), TG (β = −0.072, 95%CI: −0.111, −0.033) and HDL-C (β = −0.032, 95%CI: −0.057, −0.007).

## 4. Discussion

This study identified 60 metabolites associated with VDD in early pregnancy. In the VDD group, the down-regulated metabolites were predominantly glycerophospholipids and fatty acyls, while the up-regulated metabolites mainly consisted of various benzenoids, organic acids and derivatives, and organic oxygen compounds. Constructed with VDD-associated metabolites, PC2, which is dominated by lineolic acids and derivatives, showed a negative correlation with TC, TG, and LDL-C; PC3, mainly composed by glycerophosphocholines, exhibited an inverse relationship with TC, TG, and HDL-C. MSEA identified significant enrichment of the pantothenate and CoA biosynthesis pathway.

As for the exploration of vitamin D-related differential metabolites, in comparison with previous metabolomics studies, our findings both support and extend existing evidence. For instance, Li et al. [9] collected early pregnancy serum samples and applied a targeted metabolomics profiling method, which revealed that several fatty acyls, glycerolipids, glycerophospholipids and sterol lipids were downregulated in the VDD group. The ATBC Study in middle-aged men found that 25(OH)D levels correlate with several lipid metabolites including erucoyl sphingomyelin, eicosapentaenoate (EPA) and docosahexaenoate (DHA), and so on [13]. Also, the Hong Kong Osteoporosis Study identified correlations between 13 metabolites and 25(OH)D, while DHA and EPA had the highest correlations [14]. In patients with critical illness, Amrein et al. [15] found that as the 25(OH)D levels increased, the plasma levels of plasmalogen, lysoplasmalogen, and lysophospholipid metabolite classes also increased simultaneously, suggesting that vitamin D status is associated with glycerophospholipids and fatty acids across different populations, while the exact metabolite species involved may vary between studies. Another study using pregnant mice as the experimental model showed that the metabolite pathways of unsaturated fatty acids and glycerophospholipids were significantly enriched in the VDD diet group, indicating that vitamin D status profoundly affects the homeostasis of glycerophospholipids [16]. However, beyond these lipid-related findings, our study identified additional metabolite changes that were not reported in earlier investigations. Specifically, upregulated metabolites in the VDD group included various benzenoids, organic acids and derivatives, and organic oxygen compounds, suggesting that VDD may also disturb aromatic compound metabolism and broader energy-related pathways. To further explain these findings, we will discuss them in greater detail below, integrating evidence from the PCA and MSEA results.

In our study, glycerophosphocholines contributed most to PC3. As we mentioned before, the association between vitamin D and glycerophospholipid metabolites has been widely investigated in non-pregnant populations, with several studies providing mechanistic insights [13,14,17]. The effects of a natural vitamin D formulation on hepatocyte lipotoxicity were investigated in vitro. The results showed that the phosphatidylcholine pool’s chain length and number of double bonds changed significantly, as well as the metabolism of glycerophospholipids [18]. There are several hypotheses can be proposed to explain the finding. First, vitamin D/vitamin D receptor (VDR) activation may regulate the expression of peroxisome proliferator-activated receptor gamma (PPAR-γ). Activated PPARα promotes the expression of fatty acid transporters and oxidases, thereby accelerating fatty acid catabolism and ultimately reducing the availability of substrates for TG synthesis [19]. Second, it has been revealed that 25(OH)D inhibits the activation of the Sterol Regulatory Element-Binding Protein (SREBP), SREBP1c and SREBP2 are critical transcription factors engaged in TG and cholesterol biosynthesis, and the inhibition of their activity directly reduces the hepatic production of endogenous TG and cholesterol [20,21,22]. Additionally, vitamin D may improve the uptake and utilization of free fatty acids by modulating insulin signaling, thereby indirectly influencing hepatic TG synthesis and very-low-density lipoprotein (VLDL) secretion. The association between vitamin D and glycerophospholipids is also likely to be indirect. Previous studies have discovered associations between calcium, magnesium, phosphate, and metabolites [23,24,25]. Among them, phosphate plays a direct role in glycerophospholipid biosynthesis, while calcium and magnesium are also essential cofactors in numerous enzymatic and metabolic processes [26,27,28,29]. Moreover, decreased intestinal absorption of calcium, magnesium, and phosphate is a recognized potential consequence of vitamin D insufficiency [30,31]. Therefore, the lower glycerophospholipid levels observed in the VDD group in our study may be indirectly explained by disturbances in mineral homeostasis, such as reduced phosphate absorption leading to lower serum phosphate concentrations, together with potential contributions from altered calcium and magnesium availability.

Different from glycerophospholipids, which have been the primary focus in most studies examining associations between vitamin D status and metabolites, linoleic acid and its derivatives (contributed most to PC2) have been less frequently reported in non-pregnant population, suggesting their potential importance may be specific to pregnancy. An experimental study also found that linoleic acid levels were sharply reduced in both the plasma and liver of pregnant rats with VDD, suggesting that as an essential fatty acid, linoleic acid is more susceptible to amplified effects of both synthesis and consumption mechanisms under conditions of VDD [11]. The level of linoleic acid in the mother is very important for the growth and development of the fetus. Studies have shown an inverted U-shaped connection among the birth weight of the offspring and the mother’s consumption of linoleic acid, indicating that both insufficient and excessive intake may adversely affect birth outcomes [32]. Additionally, linoleic acid and its derivatives can influence placental function and fetal neurodevelopment by modulating inflammatory cytokines and cellular signaling pathways [33]. Given this, whether vitamin D can influence maternal health and neonatal birth outcomes by regulating linoleic acid and its metabolic pathways warrants further in-depth and systematic investigation.

This study also found elevated levels of various benzenoids, Organic acids and derivatives, and organic oxygen compounds in the VDD group. These findings suggest that vitamin D status may exert broader effects beyond classical lipid and energy metabolism pathways, potentially influencing biochemical clearance processes and host–microbiome homeostasis. For example, low vitamin D levels have been linked to downregulated expression and activity of key cytochrome P450 enzymes, which may slow the metabolism of various drugs and organic compounds, leading to their accumulation in the body [34].

MSEA revealed a significant enrichment in the pantothenate and CoA biosynthesis pathway, while unsaturated fatty acids biosynthetic pathways lost statistical significance after correction. The pantothenate and CoA biosynthesis pathway is a critical hub for energy production and biosynthesis in the human body. CoA is a necessary cofactor in several enzyme-catalyzed reactions, including fatty acid β-oxidation, triglyceride synthesis, and cholesterol biosynthesis. Vitamin D may regulate pantothenate kinase 4 through the modulation of PPARγ, thereby influencing the efficiency of the conversion of pantothenate to CoA [35]. However, previous studies based on the KEGG database differ from ours and are not entirely consistent with each other. Lasky-Su et al. [36] found a linkage between vitamin D status and the glutathione and glutamate metabolism pathways, while Wang et al. [37] identified differential metabolites concentrated in glycerophospholipid metabolism, fat digestion and absorption, and triglyceride metabolism pathways, and Mousa et al. [38] focused on the sphingolipid pathway. To validate these relationships, more pathway and enrichment analyses with bigger sample sizes are required.

Overall, there is a limited number of metabolomic studies on VDD in pregnant women, and the conclusions remain inconsistent. Additionally, existing studies have not considered the relationship between vitamin D status, metabolites and lipid profiles together. This study provides new proof of vitamin D’s beneficial effect on pregnant women’s lipid levels and offers insights into the underlying mechanisms. This study still has a number of shortcomings, though. First, because it is a cross-sectional study, causal correlations cannot be established. Including mid-pregnancy lipid data for longitudinal analysis would better assess the impact of vitamin D on lipid levels and help elucidate the underlying mechanisms. Second, the statistical power is limited by the small sample size, which may therefore limit how broadly the results can be applied, future studies with larger populations are warranted.

## 5. Conclusions

Pregnant women with VDD exhibited alterations in 60 metabolites, particularly down-regulation of glycerophospholipids and fatty acyls. A linoleic acid–dominated component (PC2) and a glycerophosphocholine-dominated component (PC3) both showed inverse correlations with maternal lipid levels. MSEA further revealed significant enrichment of the pantothenate and CoA biosynthesis pathway. This study provides new metabolomic evidence for the linkage between vitamin D and maternal lipid levels during pregnancy, along with clues about the mechanisms.

## Figures and Tables

**Figure 1 nutrients-17-03096-f001:**
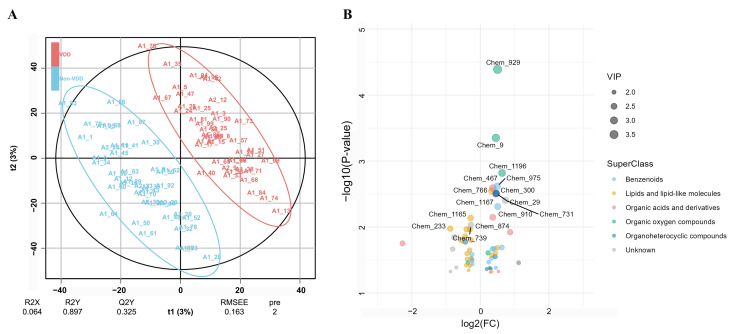
VDD-associated metabolites by the PLS-DA method. (**A**) The score plot for multivariate modeling using the PLS-DA approach reveals an overlapping of metabolome between VDD and non-VDD participants. The score plot depicts percentage of response variable explained by the first and second predictors (t1 and t2). R^2^X (predictors explained by the model), R^2^Y (grouping explained by the model), and Q^2^Y (predictive ability estimated via cross-validation) are listed on the figure and indicate overfitting. (**B**) Volcano plot depicting increased and decreased VDD-related metabolites.

**Figure 2 nutrients-17-03096-f002:**
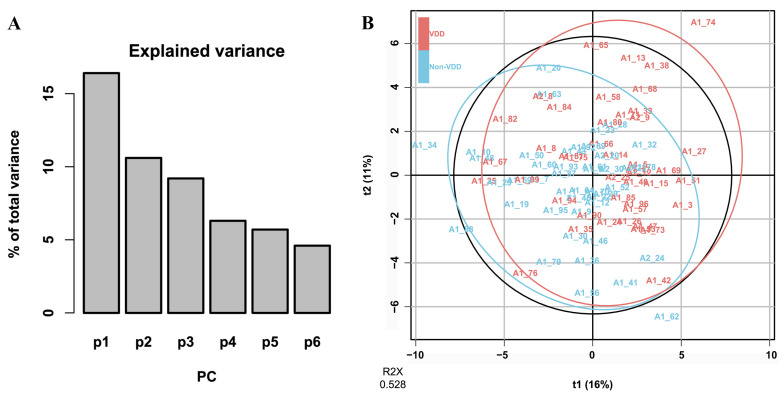
PCA results based on VDD-associated metabolites. (**A**) The proportion of total variance explained by the PCs. (**B**) The Score plot of multivariate modeling by the PCA approach.

**Figure 3 nutrients-17-03096-f003:**
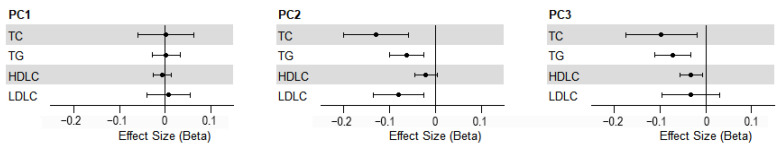
Associations between TOP 3 PCs and lipid levels. TC, total cholesterol; TG, triglycerides; HDL-C, high-density lipoprotein cholesterol; LDL-C, low-density lipoprotein cholesterol.

**Table 1 nutrients-17-03096-t001:** The distribution of the relevant characteristics between subjects with VDD and Non-VDD.

Characteristics	Non-VDD	VDD	*p*
	(*n* = 44)	(*n* = 44)	
Age, median [IQR], yr	29.00 [27.75, 31.25]	29.00 [27.00, 33.00]	0.903
BMI, median [IQR], kg/m^2^	21.18 [20.06, 22.95]	22.85 [20.98, 24.27]	0.055
Gravidity, *n* (%)			
1	18 (40.91)	22 (50.00)	0.646
2	16 (36.36)	11 (25.00)	
≥3	9 (20.45)	9 (20.45)	
Unknown	1 (2.27)	2 (4.55)	
Parity, *n* (%)			0.264
Nulliparous	25 (56.82)	23 (52.27)	
Multiparous	16 (36.36)	13 (29.55)	
Unknown	3 (6.82)	8 (18.18)	
Education level, *n* (%)			0.449
Junior high school and below	5 (11.36)	7 (15.91)	
Senior high school	7 (15.91)	12 (27.27)	
College and above	28 (63.64)	21 (47.73)	
Unknown	4 (9.09)	4 (9.09)	
25(OH)D, median [IQR], ng/mL	27.20 [22.53, 33.13]	13.96 [11.97, 16.43]	<0.001
25(OH)D_2_, median [IQR], ng/mL	0.62 [0.38, 1.00]	0.54 [0.33, 0.66]	0.251
25(OH)D_3_, median [IQR], ng/mL	26.20 [22.08, 32.45]	13.35 [11.33, 15.85]	<0.001
Detection week, median [IQR], wk	12.00 [11.43, 12.75]	12.00 [11.55, 12.71]	0.957
Detection season, *n* (%)			0.319
Spring	5 (11.36)	11 (25.00)	
Summer	12 (27.27)	8 (18.18)	
Autumn	15 (34.09)	16 (36.36)	
Winter	12 (27.27)	9 (20.45)	

Note: BMI, Body mass index.

**Table 2 nutrients-17-03096-t002:** Primary metabolites of TOP 3 PCs.

Metabolites ID	Scores	MS2_Name	Super.Class	Class	Sub.Class
PC1					
Chem_210	−0.261	Hexylresorcinol	Benzenoids	Phenols	Benzenediols
Chem_594	−0.254	Perillic_acid	Lipids and lipid-like molecules	Prenol lipids	Monoterpenoids
Chem_590	−0.251	Nadolol	Benzenoids	Tetralins	Tetralins
Chem_719	−0.248	Octenoyl-carnitine	Lipids and lipid-like molecules	Fatty Acyls	Fatty acid esters
Chem_504	−0.248	.alpha.-Hydroxymetoprolol	Benzenoids	Phenol ethers	Phenol ethers
Chem_983	−0.248	Androstan-4,6-diene-17.beta.-ol-3-one	Lipids and lipid-like molecules	Steroids and steroid derivatives	Androstane steroids
Chem_997	−0.235	Bis(2,4,6-trimethylphenyl)phosphine	Benzenoids	Benzene and substituted derivatives	Benzene and substituted derivatives
Chem_766	−0.229	Hexenoylcarnitine (Car(6:1))	Lipids and lipid-like molecules	Fatty Acyls	Fatty acid esters
Chem_1196	−0.201	3-Oxocyclobutanecarboxylic acid	Organic oxygen compounds	Organooxygen compounds	Carbonyl compounds
PC2					
Chem_788	0.313	9Z,11E,13E-Octadecatrienoic acid	Lipids and lipid-like molecules	Fatty Acyls	Lineolic acids and derivatives
Chem_1197	0.294	12,13-Dihydroxy-9Z-octadecenoic acid	Lipids and lipid-like molecules	Fatty Acyls	Fatty acids and conjugates
Chem_1109	0.278	10E,12Z-octadecadienoic acid	Lipids and lipid-like molecules	Fatty Acyls	Lineolic acids and derivatives
Chem_831	0.264	(5E,9E)-Farnesylacetone	Lipids and lipid-like molecules	Prenol lipids	Diterpenoids
Chem_1022	0.258	DG(18:2(9Z,12Z)/15:0/0:0)	Lipids and lipid-like molecules	Fatty Acyls	Lineolic acids and derivatives
Chem_1199	0.245	Lepidiumterpenoid	Lipids and lipid-like molecules	Prenol lipids	Diterpenoids
Chem_1251	0.241	Ergostane-3,6-dione	Lipids and lipid-like molecules	Steroids and steroid derivatives	Ergostane steroids
PC3					
Chem_29	−0.294	N1-Methyl-2-pyridone-5-carboxamide	Organoheterocyclic compounds	Pyridines and derivatives	Pyridinecarboxylic acids and derivatives
Chem_300	−0.294	N1-Methyl-4-pyridone-3-carboxamide	Organoheterocyclic compounds	Pyridines and derivatives	Pyridinecarboxylic acids and derivatives
Chem_256	−0.253	1-Myristoyl-sn-glycero-3-phosphocholine (LPC(14:0/0:0))	Lipids and lipid-like molecules	Glycerophospholipids	Glycerophosphocholines
Chem_1165	−0.247	PC(20:3(5Z,8Z,11Z)/14:1(9Z))	Lipids and lipid-like molecules	Glycerophospholipids	Glycerophosphocholines
Chem_694	−0.246	5-Nitro-2-toluidine	Benzenoids	Benzene and substituted derivatives	Nitrobenzenes
Chem_392	−0.235	PC(14:0/16:0)	Lipids and lipid-like molecules	Glycerophospholipids	Glycerophosphocholines
Chem_1158	−0.224	PC(16:0/16:1(9Z))	Lipids and lipid-like molecules	Glycerophospholipids	Glycerophosphocholines
Chem_785	−0.206	1,2-Di-(9Z,12Z,15Z-octadecatrienoyl)-sn-glycero-3-phosphocholine	Lipids and lipid-like molecules	Glycerophospholipids	Glycerophosphocholines
Chem_9	−0.202	Pantothenic acid	Organic oxygen compounds	Organooxygen compounds	Alcohols and polyols

**Table 3 nutrients-17-03096-t003:** Metabolite set enrichment analysis (MSEA) of VDD versus non-VDD participants.

Metabolite Set Name	Total Set Size	Hits	Statistic Q	Raw *p*	FDR
Pantothenate and CoA biosynthesis	20	3	7.876	0.0002	0.0118
Biosynthesis of unsaturated fatty acids	36	7	2.742	0.0387	0.4401
Fatty acid biosynthesis	47	5	2.780	0.0511	0.4401
Fatty acid elongation	38	1	4.314	0.0522	0.4401
Fatty acid degradation	39	1	4.314	0.0522	0.4401
Nicotinate and nicotinamide metabolism	15	5	2.957	0.0561	0.4401
Tryptophan metabolism	41	4	2.813	0.0631	0.4401
Linoleic acid metabolism	5	2	2.949	0.0774	0.4401
Butanoate metabolism	15	3	2.605	0.0907	0.4401

## Data Availability

The data presented in this study are available on request from the corresponding author due to ethical and privacy restrictions.

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
