# Peer review of "Alterations in the Metabolic and Lipid Profiles Associated with Vitamin D Deficiency in Early Pregnancy"

_nutrients, 2025, doi:10.3390/nu17193096_

Round 1
Reviewer 1 Report
Comments and Suggestions for Authors
Dear authors,
Vitamin D deficiency is a nutritional problem among pregnant women. The relationship between vitamin D status and metabolic profiles during early pregnancy is significant, given the implications of deficit for both women and offspring health.
1.Introduction: presents adequate data on the subject studied, but mentions that to date there are only 3 such studies. We ask the authors to search for articles in more databases because there are many more published studies on this subject.
- “Vitamin D deficiency (VDD) is common in pregnancy and may affect lipid metabolism and its mechanism is unclear.”: according to literature data, the mechanisms are multifaceted: disrupted adipogenesis and altered secretion of adipocytokines, systemic inflammation and oxidative stress intervention.
- Methods
2.1. Study Design and Participants
-“More detailed cohort introduction, including eligibility criteria, is available in the published paper [3]”: I don’t understand what published paper the authors are referring to. I believe that the inclusion and exclusion criteria should have been mentioned in Methods.
-The authors further present the methodology adequately.
3.Results
-A drawback of the study could be the small number of study participants
-I believe that the statistical processing is of high quality
Discussions: we suggest the authors to discuss the obtained data compared to the results of other studies.
Author Response
Comment 1: Introduction: presents adequate data on the subject studied, but mentions that to date there are only 3 such studies. We ask the authors to search for articles in more databases because there are many more published studies on this subject.
Response 1: Thank you for comments. To date, we have found only three studies in pregnant women that applied metabolomics approaches to investigate vitamin D-related metabolites (Finkelstein JL et al. [7], Li X et al. [8], and Kim M et al. [9]). Nevertheless, we fully agree with your suggestion that a broader range of research should also be considered. Therefore, we have supplemented the Introduction with additional studies in pregnant women that examined vitamin D in relation to specific metabolites, such as fatty acids and bile acids, thereby providing a more comprehensive context (Page 4, Lines 197–204):
“Another study focusing on fatty acids reported that maternal 25(OH)D levels were positively associated with plasma omega-6 fatty acids, and total polyunsaturated fatty acids, while being inversely associated with total saturated fatty acid sand total monounsaturated fatty acids at delivery [11]. In addition, a study (n = 80) in Turkey women demonstrated that VDD was associated with altered profiles of bile acids [12]. Although these studies were conducted across diverse populations and different stages of pregnancy, they collectively suggest a potential link between vitamin D status and metabolic pathways.”
Comment 2:“Vitamin D deficiency (VDD) is common in pregnancy and may affect lipid metabolism and its mechanism is unclear.”: according to literature data, the mechanisms are multifaceted: disrupted adipogenesis and altered secretion of adipocytokines, systemic inflammation and oxidative stress intervention.
Response 2: We corrected the statement in the Abstract as follows (Page 1, Lines 4-6):
“Vitamin D deficiency (VDD) is common in pregnancy and may affect lipid metabolism. The underlying mechanism are multifactorial, but most evidence so far comes from non-pregnant populations. This study aims to identify the association of metabolites and metabolic patterns with VDD in early pregnancy and to evaluate their relationships with maternal lipid profiles.”
Comment 3: Methods:
2.1. Study Design and Participants
-“More detailed cohort introduction, including eligibility criteria, is available in the published paper [3]”: I don’t understand what published paper the authors are referring to. I believe that the inclusion and exclusion criteria should have been mentioned in Methods.
-The authors further present the methodology adequately.
Response 3: Considering the repetition rate, we cited the previous publication based on ZPWC in the first version of the manuscript to simplify the description of the cohort. Thank you for your comments, we have supplemented this section at Lines 214-271, Page 4-5:
“The Zhoushan Pregnant Women Cohort (ZPWC), an ongoing prospective cohort that was started in August 2011 at Zhoushan Maternal and Child Health Care Hospital in China, served as the basis for the nested case-control research design of this investigation. Eligibility for ZPWC required participants to: (1) provide informed consent; (2) be enrolled between 8 and 12 gestational weeks; (3) complete routine perinatal examinations and plan to deliver at the study hospital; (4) be between 18 and 45 years of age; (5) have no family history of psychiatric illness. Women were excluded if they had: (1) a history of major chronic or acute diseases; (2) a history of psychiatric disorders before pregnancy; (3) threatened abortion; (4) fetal malformations or abnormal fetal growth; or (5) difficulties completing questionnaires due to cognitive impairment. During the ZPWC study, trained nurses conducted in-person interviews to gather participants' sociodemographic details, lifestyle habits, and health-related behaviors. Laboratory examination results were taken from the electronic medical record system of the hospital. Fasting blood samples in each trimester were also collected.
Comment 4: Results
-A drawback of the study could be the small number of study participants
-I believe that the statistical processing is of high quality
Response 4: We agree with the reviewer that the relatively small sample size is a limitation of our study. We have acknowledged this in the Discussion section (Page 13, Lines 582-584), and further emphasized that larger, well-designed studies are needed to validate our findings:
“Second, the small sample size and lack of diversity in racial composition limit the statistical power and may restrict the generalizability of the findings, future studies with larger populations are needed to confirm this findings.”
Comment 5: Discussions: we suggest the authors to discuss the obtained data compared to the results of other studies.
Response 5: We have expanded the Discussion section in two aspects.
(1) We incorporated more references from studies conducted in non-pregnant populations and, when comparing our findings with other vitamin D-related metabolomics studies, we provided a more detailed description of the study populations, and similarities or differences in the results (Lines 421-452 Pages 9-10):
“As for the exploration of vitamin D-related differential metabolites, in comparison with previous metabolomics studies, our findings both support and extend existing evidence. For instance, Li et al. [9] collected early pregnancy serum samples and applied a targeted metabolomics profiling method, which revealed that several fatty acyls, glycerolipids, glycerophospholipids and sterol lipids were downregulated in the VDD group. The ATBC Study in middle-aged men found that 25(OH)D levels correlate with several lipid metabolites including erucoyl sphingomyelin, eicosapentaenoate (EPA) and docosahexaenoate (DHA), and so on [12]. Also, the Hong Kong Osteoporosis Study identified correlations between 13 metabolites and 25(OH)D, while DHA and EPA had the highest correlations [13]. In patients with critical illness, Amrein et al. [14] found that as the 25(OH)D levels increased, the plasma levels of plasmalogen, lysoplasmalogen, and lysophospholipid metabolite classes also increased simultaneously, suggesting that vitamin D status is associated with glycerophospholipids and fatty acids across different populations, while the exact metabolite species involved may vary across studies. Another study using pregnant mice as the experimental model showed that the metabolite pathways of unsaturated fatty acids and glycerophospholipids were significantly enriched in the VDD diet group, indicating that vitamin D status profoundly affects the homeostasis of glycerophospholipids [15]. However, beyond these lipid-related findings, our study identified additional metabolite changes that were not reported in earlier investigations. Specifically, upregulated metabolites in the VDD group included various benzenoids, organic acids and derivatives, and organic oxygen compounds, suggesting that VDD may also disturb aromatic compound metabolism and broader energy-related pathways. To further explain these findings, we will discuss them in greater detail below, integrating evidence from the PCA and MSEA results.”
(2) We additionally discussed the hypothesis that vitamin D may indirectly influence glycerophospholipid metabolism through its regulatory effects on mineral homeostasis, including calcium, phosphate, and magnesium (Lines 472-508, Pages 10-11):
“The association between vitamin D and glycerophospholipids is also likely to be indirect. Previous studies have discovered associations between calcium, magnesium, phosphate, and metabolites [23–25] Among them, phosphate plays a direct role in glycerophospholipid biosynthesis, while calcium and magnesium are also essential cofactors in numerous enzymatic and metabolic processes [26–29]. Moreover, decreased intestinal absorption of calcium, magnesium, and phosphate is a recognized potential consequence of vitamin D insufficiency [30,31]. Therefore, the lower glycerophospholipid levels observed in the VDD group in our study may be indirectly explained by disturbances in mineral homeostasis, such as reduced phosphate absorption leading to lower serum phosphate concentrations, together with potential contributions from altered calcium and magnesium availability.”
Reviewer 2 Report
Comments and Suggestions for Authors
This well-designed study reports a metabolomic assessment in plasma samples collected from pregnant women in the 8 to 14 weeks of gestation. The study used a paired nested approach based on two groups – those that were vitamin D deficient and those that were assessed as having adequate vitamin D status. This nested approach was a good strategy for diminishing the many variables when comparing those with vitamin D deficiency and those with adequate vitamin D status. The many metabolite differences between the two groups indicated the profound influence of vitamin D deficiency on metabolism, particularly of lipid metabolism.
There is one curious uncertainty in this study and other studies referred to. That uncertainty is whether the many metabolic differences found in vitamin D deficient pregnant women are also found in non-pregnant women, or indeed in men. All the studies referred to in other human pregnancy vitamin D status analyses and in animal studies were all analyses during pregnancy. Because vitamin D deficiency is a common abnormality in non-pregnant women and in men, it is important to know whether the changes reported in this manuscript are specific for vitamin D-deficient pregnant women or whether they are changes associated with vitamin D deficiency in non-pregnant women and in men. It would be important for this uncertainty to be resolved in the Discussion section of the manuscript. If these changes are only found in pregnant women then the importance, particularly for fetal development, becomes highly relevant. However, if these are changes seen in human vitamin D deficiency independent of pregnancy, their significance in pregnancy becomes less compelling.
One small error that needs correcting is in the first line of the Introduction where it is stated: “ Vitamin D is a steroid hormone with essential physiological functions.” Vitamin D itself is not a steroid hormone, but is the precursor of a steroid hormone, 1,25-dihydroxyvitamin D. It is misleading to call vitamin D itself, a steroid hormone.
Author Response
Comment 1: There is one curious uncertainty in this study and other studies referred to. That uncertainty is whether the many metabolic differences found in vitamin D deficient pregnant women are also found in non-pregnant women, or indeed in men. All the studies referred to in other human pregnancy vitamin D status analyses and in animal studies were all analyses during pregnancy. Because vitamin D deficiency is a common abnormality in non-pregnant women and in men, it is important to know whether the changes reported in this manuscript are specific for vitamin D-deficient pregnant women or whether they are changes associated with vitamin D deficiency in non-pregnant women and in men. It would be important for this uncertainty to be resolved in the Discussion section of the manuscript. If these changes are only found in pregnant women then the importance, particularly for fetal development, becomes highly relevant. However, if these are changes seen in human vitamin D deficiency independent of pregnancy, their significance in pregnancy becomes less compelling.
Response 1: Thanks for your creative comments. We searched for studies in non-pregnant populations and found several report associations between serum 25(OH)D and metabolites, especially lipid / glycerophospholipid-related metabolites. For instance, the ATBC Study in middle-aged men found that 25(OH)D levels correlate with several lipid metabolites including erucoyl sphingomyelin, eicosapentaenoate (EPA), docosahexaenoate (DHA) and so on [1]. Another study based on older adults found two short chain fatty acid concentrations were higher in the vitamin D insufficient participants, and eleven glycerophospholipid concentrations were lower [2]. Also, the Hong Kong Osteoporosis Study found 13 metabolites were highly correlated with 25(OH)D, DHA and EPA had the highest correlations [3].
In summary, evidence from non-pregnant populations indicates that vitamin D status is associated with lipid metabolites, particularly glycerophospholipids and fatty acids, which aligns with our findings. Nevertheless, the specific differential metabolites identified in these studies do not fully overlap with those observed in our cohort. Notably, multiple linoleic acids and their derivatives, which dominate PC2 in our analysis, have been less frequently reported in previous work, and we highlight their potential roles in fetal growth and development. In response to your suggestion, we have expanded and refined this discussion (Lines 510-526, Page 11):
“Different from glycerophospholipids, which have been the primary focus in most studies examining associations between vitamin D status and metabolites, linoleic acid and its derivatives (contributed most to PC2) have been less frequently reported in non-pregnant population, suggesting their potential importance may be specific to pregnancy. An experimental study also found that linoleic acid levels were sharply reduced in both the plasma and liver of pregnant rats with VDD, suggesting that as an essential fatty acid, linoleic acid is more susceptible to amplified effects of both synthesis and consumption mechanisms under conditions of VDD [11]. The level of linoleic acid in the mother is very important for the growth and development of the fetus. Studies have shown an inverted U-shaped connection among the birth weight of the offspring and the mother's consumption of linoleic acid, indicating that both insufficient and excessive intake may adversely affect birth outcomes [32]. Additionally, linoleic acid and its derivatives can influence placental function and fetal neurodevelopment by modulating inflammatory cytokines and cellular signaling pathways [33]. Given this, whether vitamin D can influence maternal health and neonatal birth outcomes by regulating linoleic acid and its metabolic pathways warrants further in-depth and systematic investigation.”
Comment 2: One small error that needs correcting is in the first line of the Introduction where it is stated: “Vitamin D is a steroid hormone with essential physiological functions.” Vitamin D itself is not a steroid hormone, but is the precursor of a steroid hormone, 1,25-dihydroxyvitamin D. It is misleading to call vitamin D itself, a steroid hormone.
Response 2: Thank you. we have rewritten this sentence (Lines 108-111, Page 3):
“Vitamin D is a fat-soluble vitamin with diverse physiological functions beyond skeletal health, including roles in immune regulation and metabolic processes [1].”
Reviewer 3 Report
Comments and Suggestions for Authors
I understand that the authors wished to identify metabolites that are associated with low serum levels of vitamin D. But they should have also measured serum levels of calcium, magnesium, and phosphate. A stronger statistical association between the latter and the identified metabolites would indicate an indirect relationship with low vitamin D levels. In other words, low vitamin D levels hinder the absorption of calcium, magnesium, and phosphate from food passing through the intestines. As a result, these minerals have lower serum levels.
I believe that the low glycerophospholipid levels in this study are due to low serum phosphate levels, since phosphate is required to synthesize glycerophospholipids. The relationship with low vitamin D levels is probably indirect.
Corrections:
Abstract, second line: replace “it’s” with “its”. It would also be better to start a new sentence: “… may affect lipid metabolism. Its mechanism is unclear.”
“selected into the study” – replace with: “selected for the study”
Page 6 – “Scores plot of multivariate modeling by the PLS-DA approach reveals highlight overlapping overall metabolome between VDD and non-VDD participants.”
Replace with: “The score plot for multivariate modelling using the PLS-DA approach reveals an overlapping of metabolome between VDD and non-VDD participants.”
Page 10 – “providing new metabolomic evidence for explaining the protective effect of vitamin D”
Replace with “providing new metabolomic evidence for the protective effect of vitamin D”
“A few of existing studies have suggested”
Replace with: “A few existing studies have suggested”
Comments on the Quality of English LanguageThe writing style is awkward and could be improved. Please see my suggested corrections.
Author Response
Comment 1: I understand that the authors wished to identify metabolites that are associated with low serum levels of vitamin D. But they should have also measured serum levels of calcium, magnesium, and phosphate. A stronger statistical association between the latter and the identified metabolites would indicate an indirect relationship with low vitamin D levels. In other words, low vitamin D levels hinder the absorption of calcium, magnesium, and phosphate from food passing through the intestines. As a result, these minerals have lower serum levels.
I believe that the low glycerophospholipid levels in this study are due to low serum phosphate levels, since phosphate is required to synthesize glycerophospholipids. The relationship with low vitamin D levels is probably indirect.
Response 1: We appreciate the reviewer’s insightful suggestion. We have revised the Discussion to acknowledge this important point (Pages 10-11, Lines 472-508), and we will include measurements of calcium, magnesium, and phosphate in future studies to further clarify this mechanism:
“The association between vitamin D and glycerophospholipids is also likely to be indirect. Previous studies have discovered associations between calcium, magnesium, phosphate, and metabolites [23–25] Among them, phosphate plays a direct role in glycerophospholipid biosynthesis, while calcium and magnesium are also essential cofactors in numerous enzymatic and metabolic processes [26–29]. Moreover, decreased intestinal absorption of calcium, magnesium, and phosphate is a recognized potential consequence of vitamin D insufficiency [30,31]. Therefore, the lower glycerophospholipid levels observed in the VDD group in our study may be indirectly explained by disturbances in mineral homeostasis, such as reduced phosphate absorption leading to lower serum phosphate concentrations, together with potential contributions from altered calcium and magnesium availability.”
Comment 2: Corrections:
Abstract, second line: replace “it’s” with “its”. It would also be better to start a new sentence: “… may affect lipid metabolism. Its mechanism is unclear.”
“selected into the study” – replace with: “selected for the study”
Page 6– “Scores plot of multivariate modeling by the PLS-DA approach reveals highlight overlapping overall metabolome between VDD and non-VDD participants.”
Replace with: “The score plot for multivariate modelling using the PLS-DA approach reveals an overlapping of metabolome between VDD and non-VDD participants.”
Page 10 – “providing new metabolomic evidence for explaining the protective effect of vitamin D”
Replace with “providing new metabolomic evidence for the protective effect of vitamin D”
“A few of existing studies have suggested”
Replace with: “A few existing studies have suggested”
Response 2: We corrected all points that you mentioned. Thanks.